



# Land-use changes influence soil bacterial communities in a meadow grassland in Northeast China

Chengyou Cao[1], Ying Zhang[1], Wei qian[1], Caiping Liang[1], Congmin Wang[1], Shuang Tao[1]

[1]College of Life and Health Sciences, Northeastern University, Shenyang 110169, P R China

*Correspondence to*: Chengyou Cao (caochengyou@mail.neu.edu.cn)

**Abstract.** The conversion of natural grassland into agricultural fields is an intensive anthropogenic perturbation commonly occurring in semi-arid regions, and this perturbation strongly affects soil microbiota. In this study, the influences of land-use conversion on the soil properties and bacterial communities in Horqin Grasslands in Northeast China were assessed. This study aimed to investigate (1) how the abundances of soil
bacteria changed across land-use types; (2) how the structure of soil bacterial community was altered in each land-use type; and (3) how these variations were correlated with soil physical and chemical properties. The variations in diversities and compositions of bacterial communities and relative abundance of dominant taxa were detected in four distinct land-use systems, namely, natural meadow grassland, paddy field, upland field, and poplar plantation, through high-throughput Illumina MiSeq sequencing technique. Results indicated that
land-use changes primarily affected soil physical and chemical properties and bacterial community structure. Soil properties, namely, organic matter, pH, total N, total P, available N and P, and microbial biomass C, N, and P, influenced the bacterial community structure. The dominant phyla and genera were almost the same among the land-use types, but their relative abundances were significantly different. The effects of land-use changes on the structure of soil bacterial communities were more quantitative than qualitative.

**Keywords** Illumina MiSeq sequencing; soil microbial diversity; composition of microbial community; Soil factors; 16S rRNA



# 1 Introduction

Land-use conversion, habitat destruction, climate change, and other intensive anthropogenic activities strongly alter natural ecosystems (Pabst et al., 2013). Over 38% of the entire natural landscapes worldwide are converted to managed systems (Holland et al., 2016). Thus, response processes should be elucidated to predict the development and sustainability of environmental services of ecosystems. Arid and semi-arid areas account for 30% of the total land area worldwide (Ran et al., 2014). These areas are also known for their ecological fragility and resource specificity. The conversion of natural grassland into agricultural fields is a dominant human perturbation commonly occurring in semi-arid areas. In the western part of Northeast China, large grassland areas have been converted to upland field or paddy field because of high income from farmlands. In the coming decades, many semi-arid grassland areas in Northeast China are expected to be converted for agriculture use. This conversion can cause progressive and cumulative soil disturbances and thus is considered a key influencing factor that affects land desertification development, reduces biodiversity, and alters ecological processes in terrestrial ecosystems (Sala et al., 2000). Intensive soil perturbations can also change intrinsic soil properties by promoting soil nutrient cycles and modifying soil physical, chemical, and microbiological properties and structures of dwelling microbial communities in local ecosystems (Lauber et al., 2008; Lin et al., 2011; Kuramae et al., 2012; Lauber et al., 2013). Therefore, land-use or land-cover change is an important factor affecting soil quality. Soil microbial diversity and community structure function as sensitive indicators of soil health and quality and hence rapidly respond to land-use conversions (He et al., 2008).

The effects of land-use conversion on soil physical, chemical, and microbiological properties have been comprehensively investigated. Land-use conversion can elicit significant and long-term effects on the moisture, texture, aeration, pH, nutrient status, microbial biomass, and enzymatic activities of soil largely because of the changes in plant community composition and different management practices across land-use types (Murty et al., 2002; Nishimura et al., 2008; Rahman et al., 2008; Wallenius et al., 2011). However, the effects of land-use conversion on soil biology have yet to be sufficiently assessed. Although land-use conversion has been



confirmed to affect the structure of soil microbial communities significantly (Steenwerth et al., 2002; Johnson et al., 2003; Lauber et al., 2008), changes in the abundance of soil-specific taxonomic groups modified during land-use conversion and alterations in plant cover and soil properties remain poorly understood. Soil microbial communities are structured by climate, geology, land use, and soil physicochemical factors, such as soil organic matter, porosity, and pH (Thomson et al., 2010; Griffiths et al., 2011; Hartmann et al., 2012; Thomoson et al., 2015). Land-use conversion may lead to unfavorable modification of several environmental variables, which can indirectly or directly affect soil microbial diversity. Therefore, the consequences of changes in soil microbial diversity and community structure during land-use conversion should be investigated to provide relevant information on microbial responses to changes in soil environment and interactions between soil microbial communities and soil physicochemical properties.

Culture-independent techniques, such as phospholipid fatty acid biomarker, polymerase chain reaction (PCR)-denaturing gradient gel electrophoresis, clone libraries, fluorescence in situ hybridization, enterobacterial repetitive intergenic consensus, and terminal restriction fragment length polymorphism, have been widely employed to examine soil microbial community changes as a consequence of land cover change, soil pollution, and fertilization (Baath et al., 2003; He et al., 2007; Berg et al., 2012; Suleiman, et al., 2013; Thomson et al., 2015). However, these techniques yield unsatisfactory performances in terms of the accurate detection of the changes in ecologically important soil microbial groups under different soil environmental conditions because of resolution limit, fingerprint complexity, and highly abundant indigenous microbes in soil (Bruce et al., 2000; Quince et al., 2009). Conversely, DNA-characterization-based high-throughput methodologies, such as sequencing and subsequent sequence alignment, can directly detect microbial community composition and variations in species with low richness (Shi et al., 2014; Zhao et al., 2014). These methodologies have also been widely adopted to assess the genetic structure and diversity of microbial communities in various environments (Quince et al., 2009; Gołębiewsk et al., 2014; Liu et al., 2014; Portune et al., 2014; Ren et al., 2014; Singh et al., 2014; Thomson et al., 2015).

Considering the increasing number of reclamation activities in native meadow grasslands in Northeast



China, we applied Illumina MiSeq sequencing to investigate the changes in the diversities, compositions, and relative abundances of the dominant taxa of soil bacterial communities in native meadow grassland and adjacent upland field, paddy field, and woodland habitats. We then classified the microbial species and compared their structures among different land-use types. We also analyzed the correlation of specific microbial phyla with soil physicochemical properties. Thus, we could obtain relevant data to enhance our understanding on the effects of native grassland cultivation on soil microbiota, and to provide useful information for the future development of sustainable use of fragile ecosystem services. We also statistically analyzed our data to evaluate the differences in the structures of soil bacterial communities among different land-use types (namely, native meadow grassland, upland field, paddy field, and woodland) and to determine their correlations with soil physicochemical properties. We hypothesized that soil bacterial communities respond to land-use conversion, and these responses are related to variations in soil properties induced by different managements during changes. This study aimed (1) to determine the changes in the abundances of soil bacteria across land-use types, (2) to identify the alterations in the structure of soil microbial communities in each land-use type, and (3) to observe the responses of specific microbial groups to land-use changes and the correlation of these variations with soil physicochemical properties.

## 2 Materials and methods

### 2.1 Study location and site description

This study was conducted at the Wulanaodu Experimental Station of Desertification (43°02' N, 119°39' E) under the Institute of Applied Ecology, Chinese Academy of Sciences. The station is located in the western part of the Horqin Grassland in Northeast China. The Wulanaodu region is located in the temperate zone and thus has a continental semi-arid monsoon climate with mean annual precipitation of approximately 284.4 mm. In addition, 70%−80% of the precipitation occurs from May to September. The annual average temperature and wind velocity in the area are 6.3 ℃ and 4.4 m s$^{-1}$, respectively. The soils are classified as alkali meadow soil.

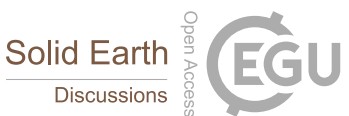

The landscape is characterized by a mosaic of gently undulating sand dune and interdune meadow grassland (Zhang et al., 2016). The original vegetation belongs to the Mongolian flora, which mainly includes *Aneurolepidium chinense*, *Chenopodium acuminatum*, *Bassia dasyphylla*, *Lespedeza davurica*, *Thermopsis lanceolata*, *Achnatherum cristatum*, *Arundinella hirta*, *Caragana microphylla*, *Salix mongolica*, *Spodiopogon sibircus*, and *Artemisia frigida*. Most of the herbaceous plant species in this region are distributed in the meadow grassland. Furthermore, a large native meadow grassland lies in the Wulanaodu region and supports the production of local animal husbandry. The meadow grassland are typically used as clipping pastures, which are enclosed during the growing season (from April to late September) and openly grazed after the grasses are harvested (Zhang et al., 2015). In the recent decades, the grasslands have been seriously degraded because of climate change, overgrazing, and aridification of habitat (Cao et al., 2006), and a vast area of the degraded meadow grasslands was converted to farmlands. At present, a mosaic of grassland, paddy field, upland field, and woodland can be observed in the Wulanaodu region, and these ecosystems can be used to study the response of soil microbial community structure to land-use conversion.

## 2.2 Experimental design and soil sampling

The soils were sampled in May 2016. Adjacent native meadow grassland, 26-year paddy field (*Oryza sativa* L.), 50-year upland field (*Zea mays* L.), and 34-year poplar plantation (designated as NMG, PF, UF, and PP, respectively), were selected as experimental sites. PF, UF, and PP were all converted from NMG, and thus the soil types were the same. NMG was openly grazed except for growing season. At PF and UF sites, urea, $(NH_4)_2HPO_4$, and KCl were annually applied during maize-seeding, rice-transplanting, and crop growth. The averages of height and diameter at breast height of trees were 18.5 m and 20.6 cm at PP site, respectively. In each site, three 30 m $\times$ 30 m plots were installed for sampling, and each plot was 300 m away from the other plots. Approximately 0–10 cm soil samples were collected. All the samples were sieved in the field using a 2 mm screen. Half of each sample was immediately frozen at −80 ℃ for DNA extraction, while the other half sample was used for the analysis of the chemical and physical properties.

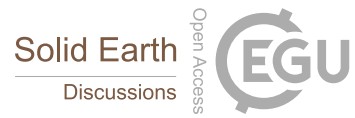

## 2.3 Soil physicochemical property and microbial biomass

Soil water content was determined gravimetrically by drying the soil at 105 ℃ for 24 h. In the laboratory, the air-dried soil was ground to pass through a 2.0 mm mesh and was then analyzed for electrical conductivity (EC; 1:5, soil: water ratio), pH (1:2.5, soil: water ratio), total K, and available N, P, and K. The soil organic matter (SOM), total N, and total P of the samples ground to pass through a 0.25 mm mesh were analyzed. Subsequently, the amount of organic matter, total N, total P, available N, and available P were analyzed through the $K_2Cr_2O_7$-$H_2SO_4$ oxidation method, semimicro-Kjedahl digestion method, acid-digestion molybdate colorimetric method, alkaline diffusion method, and molybdate ascorbic acid method (in 0.5 M $NaHCO_3$), respectively. All the analyses were performed according to the procedures described in ISSCAS (1978). The C and N in the soil microbial biomass were estimated through the chloroform fumigation-incubation method (Jenkinson and Powlson, 1976). The P in the microbial biomass was analyzed according to methods described by Brookes et al. (1982).

## 2.4 DNA extraction, PCR amplification, and 16S rDNA sequencing

Microbial DNA was extracted from 0.3 g of fresh soil sample using the Soil DNA Extraction kit (Tiangen Biotech, China) according to the manufacturer's instructions. The 16S rRNA genes were amplified using the barcode primers 341F and 805R (Hugerth et al., 2014). The 50 μL PCR mixture contained 5 μL of 10 ×reaction buffer, 10 ng of template DNA, 0.5 μL of 10 mM dNTPs, 0.5 μL of 50 μM each primer, and 0.5 μL of 5 U·μL$^{-1}$ Plantium *Taq* polymerase. PCR was performed using the following program: 94 ℃ for 3 min; 5 cycles of 94 ℃ for 30 s, 45 ℃ for 20 s, 72 ℃ for 30 s; 20 cycles of 94 ℃ for 20 s, 55 ℃ for 20 s, 72 ℃ for 30 s; and a final extension at 72 ℃ for 10 min. The PCR products were purified using the agarose gel extraction kit (Tiangen Biotech, China) according to the manufacturer's instructions and then quantified using a Qubit 2.0 fluorometer. The purified PCR products of all the samples were mixed in equal mole amounts and sequenced on an Illumina Miseq platform (Genewiz Biological Technology & Services Co., Ltd, Suzhou, China) according to the standard protocols.




**2.5 Processing and analyzing of sequencing data**

Raw sequence files were analyzed and quality-filtered using the Quantitative Insights Into Microbial Ecology (QIIME version 1.17, http://qiime.org). Reads shorter than 200 bp and average quality scores < 25 were discarded. The chimeric sequences were then identified and removed using the UCHIME software (http://drive5.com/uchime). The operational taxonomic units (OTUs) with 97% similarity cutoff were clustered using UPARSE (version 7.1, http://drive5.com/uparse). The representative sequences of each OTU were taxonomically classified using the ribosomal database project (RDP) naive Bayesian 16S rRNA Classifier (Wang et al., 2007), which assigns the complete taxonomic information from domain to species to each sequence in the database with 80% taxonomy confidence and e-value of 0.001. OTU richness analysis was conducted using Mothur software (https://www.mothur.org, version 1.21.1), and the alpha diversity indices, including the Shannon-Wiener index, Chao's species richness estimator (Chao), and abundance-based coverage estimator (ACE) were calculated (Schloss et al., 2009). Heat map analysis was performed using MeV version 4.2. The Bray–Curtis dissimilarity measure was performed to calculate the between samples similarity matrix, and principal coordinate analysis (PCoA) was performed to compare and visualize the similarities among the soil samples. All raw sequences were submitted to the NCBI Sequence Read Archive (SRA) under the accession number SRR5083094- SRR5083105.

ANOVA and multiple comparisons were performed to determine the differences among the land-use types. All the statistical analyses were performed using the SPSS software package (version 13.0, SPSS Co., Ltd, Chicago, USA). A difference at a *P* < 0.05 was considered to be statistically significant.

*2.6. Canonical correspondence analysis*

Canonical correspondence analysis (CCA) was performed using CANOCO 4.5 (Biometrics Wageningen, The Netherlands) to determine which among the soil chemical properties have the most significant effect on the composition of soil microbial community. The correlations of the soil parameters were examined using Monte Carlo permutation, and the figures were generated by CanoDraw 4.0 (Biometris-Plant Research

International, Wageningen, Netherlands).

# 3. Results

## 3.1 Soil physical and chemical parameters

The physicochemical properties and microbial biomass (C, N, and P) of the soil varied according to land use.

Significant differences in the soil moisture, pH, EC, SOM, total N, available P, and microbial C, N, and P among the land use types were observed (Table 1, $P < 0.05$). Meanwhile, no significant difference in soil total P and available N among the samples was observed. The conversions from native meadow grassland to UF, PF, or PP significantly decreased the soil pH and increased the EC and available P. The SOM, total N, and microbial C were lower in UF and PF, but higher in PP than those in NMG, respectively, that is, PP > NMG > UF > PF.

## 3.2 Sequencing results and diversity indices

A total of 646127 valid sequences and 87428 OTUs were obtained from the 12 samples through MiSeq sequencing analysis. Each sample contained 40441 to 66811 reads and had different phylogenetic OTUs ranging from 6114 to 8744 using a 3% nucleotide cutoff (Table 2). In addition, the indices of the alpha diversity, including species richness (Chao), Shannon-Wienner index, coverage, and evenness (ACE), were also

calculated to estimate the species richness and biodiversity in the 12 samples. Significant differences in the mean values of ACE, Chao, and Shannon-Wienner index among the land-use types were observed (Table 2, $P < 0.05$). The average values of the ACE and Chao in PF were significantly lower than those in the NMG, UF, and PP, while its value of Shannon-Wienner index was higher than those in NMG and PP. The coverage index, ranged from 0.93 to 0.96, indicated that significant differences among the samples were absent. PCoA was

performed according to the Bray–Curtis distance matrix (Fig. 1). The PCoA plot showed that the microbial communities from the NMG and PP grouped together, and UF1-UF3 and PF1-PF3 separately formed other groups. These findings indicated that land-use type can alter the soil microbial communities, but it is not the only determinant.

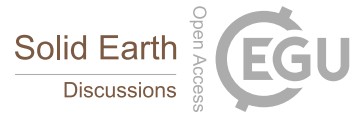

## 3.3 Taxonomic profiles

The representative sequences of each OTU were assigned using the RDP classifier at the phylum, family, genus, and species level. The OTUs were classified into 46 different phyla, 416 orders, 776 families, 1289 genera or 1801 species. Out of the 46 bacterial phyla, 13 dominant phyla with relative abundance of >1% were detected

and they include *Proteobacteria*, *Acidobacteria*, *Actinobacteria*, *Bacteroidetes*, *Chloroflexi*, *Gemmatimonadetes*, *Firmicutes*, *Planctomycetes*, *Verrucomicrobia*, *Nitrospirae*, *Thaumarchaeota*, *Chlorobi*, and candidate_division_TM7 in all the samples, which accounted for 92.48% to 97.68% of the total OTUs in the different samples. The average relative abundances of the top 30 phyla of the samples are shown in Fig. 2. The relative abundance of *Proteobacteria* ranged from 29.13% in PP to 30.50% in MS was considered to be

the dominant group in all the samples. *Acidobacteria*, *Actinobacteria*, *Bacteroidetes*, and *Chloroflexi* had higher relative abundances as well, and they ranged from 10.41% to 31.79%, 8.46% to 16.14%, 7.24% to 12.56%, and 5.37% to 16.63%, respectively. Four subdivisions of *Proteobacteria* (α, β, γ, and δ-*Proteobacteria*) were identified in all the libraries. Significant differences in relative abundance of each subdivision among the land use types were found (Fig. 3, $P < 0.05$), indicating the variations in the *Proteobacteria* phylum after land-use

conversion. α-*Proteobacteria* had the highest relative abundance among the four subdivisions in all the samples, and accounted for the 11.14%−13.82% of the total reads in the NMG and PP and 6.45%−9.78% of the total reads in the PF and UF samples (Fig. 3).

## 3.4 Core genera

Soil dominant bacterial phyla in the soil samples were compared with one another at the genus levels to further

reveal the changes in the soil microbial communities among the land-use types. In total, 1289 genera were identified in the 12 samples. The highest and lowest genera number were found in the PF (1289) and PP (890) samples, respectively. Most of genera were commonly distributed in all the samples, however, the relative abundances of most dominant genera were significantly different among the samples, and thus they reflected differences among the structures of the soil microbial community (Table 3). The dominant genera were selected

and then used to draw a histogram based on their relative abundances (Fig. 4) and depict the distribution patterns of the bacterial genera among the land-use types. The dominant genera among land-use types had similar distribution patterns, as indicated in the Fig. 4. This finding suggested that the land-use conversion of NGM have no distinct effect on the dominant genus compositions of the soil bacterial community. Among these enriched genera, *Acidobacteria*, *Blastocatella*, *Sphingomonas*, and *Bryobacter* were dominant in all the samples. However, the conversion from NMG to UF, PF, and PP significantly increased the relative abundances of some specific genera (>1%), which then became the new dominant taxa, specifically, *Marmoricola*, *Ferruginibacter*, and *Anaerolinea* in PF, *Arthrobacter*, *Nocardioides*, *Massilia*, *Adhaeribacter*, *Flavisolibacter*, and *Rubrobacter* in PF, and *Chryseolinea* in PP, which induced the structural shift of the soil microbial community at the genus level.

### 3.5 Relationship between the composition of microbial community and soil environment factors

In this study, Canonical correspondence analysis (CCA) was conducted to evaluate the relationship between the compositions of the dominant phyla or genera and selected soil properties (soil moisture, electrical conductivity, pH, organic matter, total N, P, available N and P, microbial biomass C, N, and P). The results are summarized in Fig. 5. The CCA plots based on the dominant phylum and genera were nearly identical. The overall structures of the dominant phyla or genera in different land-use samples were significantly linked to the selected soil properties. Through CCA ordination, all the samples were classified into three groups. Particularly, NMG and PP samples were grouped into a distinct cluster, while UF and PF formed the other two clusters. The main factors that grouped the NMG and PP samples were pH, organic matter, total N, and microbial biomass C. Meanwhile, cluster UF appeared to be mostly influenced by total P, available P, microbial biomass P, and electrical conductivity. Cluster PF in turn was significantly affected by soil moisture, available N, and microbial biomass N (Fig. 5).

## 4 Discussion

The purpose of present study was to evaluate the effects of land-use conversion on the composition and structure





of the soil bacterial communities and explore the relationship between the physicochemical parameters of the soil and specific phyla in the Horqin region. In this regard, UF, PP, and PF were formerly NMG, which is one of the dominant land-use type in the Horqin Sandy Land. Land-use conversion can be regarded as the determinant that can alter soil properties because the four sites have the same soil type. Consistent with studies

in other areas (Hartman et al., 2008; Wakelin et al., 2008), our results showed significant differences among the structures of the soil bacterial communities in the land-use types. These differences were related to the changes in the physicochemical properties of the soil (especially in pH and nutrient contents) and thus suggested that the soil properties were significant factors that shape the bacterial community structure. During the conversion from NMG to UF, PF, and PP, the vegetation cover and soil properties were changed according to

how the land conversion was managed. Variations in the environment due to land-use conversion have a direct influence on the microbial community inhabiting the soil (Zhang et al., 2014; Mendes et al., 2015). In the Horqin region, the land-use conversion of NMGs usually involves plowing and harrowing of the soil and cultivation of the crops or planting of trees. This process dramatically altered the soil physicochemical and microbiological properties by destroying the vegetation and increasing the rates of organic matter decomposition (Su et al.,

2004). Previous studies reported that the changes in the soil properties are attributed to cultivation and fertilization of the soil (He et al., 2007; Wang et al., 2012; Shen et al., 2010). Furthermore, the changes in the soils may involve the shifts in nutrient contents and soil quality, as the natural grassland or forest plantation areas are expected to accumulate litter deposits, which are relatively more recalcitrant than the crop debris deposited in agricultural areas (Lauber et al., 2008). In an agricultural system, a low input of organic matter,

variations in temperature and precipitation, and management of soils were reported to be related to soil properties and abundance of organisms (Lauber et al., 2013; Hartmann et al., 2015).

The PCoA and CCA analyses showed the differences among the structures of the soil microbial communities during the land-use conversion of NMG. CCA analysis performed on the soil environmental factors and relative abundances of the dominant taxa indicated that the changes in the environment due to land-

use conversion have direct and significant impacts on the community of microorganisms inhabiting the soil.





Soil P contents, such as total P, available P, and microbial biomass P were the most important factors for the microbial community structure in the UF sites. By contrast, the formation of the soil microbial community structure in the PF sites were significantly related to soil moisture and available N, as well as microbial biomass N (Fig. 5). However, NMG and PP had similar microbial community structures, which were mainly affected

by pH, organic matter, total N, and microbial biomass C. Our results demonstrated that pH was an important factor that shaped the structure of the soil microbial community. Some studies showed that the soil pH is generally related to the compositions and structures of the soil microbial communities across a geographic scale (Högberg et al., 2007; Jesus et al., 2009; Fierer and Jackson, 2006). Mendes et al. (2015) studied the effects of land-use system on soil bacterial communities in the southeastern region of the Amazon, and they confirmed

that pH is correlated with the structure formation of bacterial community during land-use alternations. Wakelin et al. (2008) also found that pH influences the structural composition and function capacity of the soil bacterial communities in Australian agricultural soils. The potential causes of this correlation are the following: (1) integration of pH with other soil physicochemical and microbiological properties, (2) a slight variation in the soil pH could expose microbes to stress, that is, the sensitivity of the microbial cell to environmental change

(Fierer and Jackson, 2006). Some studies indicated that several parameters other than pH and elements (e.g., Mg, Al, and Cd) are also correlated to specific bacterial groups and structure alternation of the microbial communities (Pan et al., 2014; Jesus et al., 2009; Mendes et al., 2015).

The vegetation cover might affect the diversity and structure of the soil microbial community because the differences in plant community compositions can contribute to changes in litter quality and quantity, which

then alter the content and cycling processes of soil nutrients (Miki et al., 2010). In this study, the four sites differed in terms of vegetation cover. Therefore, the differences among the soil properties and structures of the microbial communities in different land-use types can be expected. In the agricultural sites of our study (UF and PF), the cultivation of rice and corn was performed through conventional management, while in NMG and PP, the cultivation was performed through no-tillage management. The long-term cultivation might affect the

whole soil community. Souza et al. (2013) showed that the major differences among the soil biodiversity and





microbial community structures were associated with tillage management. Differences among the soil structures induced by different tillage systems and land-use conversion influence the structures of microbial communities (Sessitsch et al., 2001; Peixoto et al., 2006). Considering that the four land-use types have the same soil type, and the sampling sites were adjacent to each other, we can assume that land-use conversion is a principal driving force that alters the soil properties and microbial community structures. Jangid et al. (2011) investigated the rRNA gene diversity of soil bacteria at two successional gradients with different vegetation, and they found that bacterial diversity remained unchanged in both gradients; their results confirmed that land-use history was the main determinant of the species composition in the soil microbial communities rather than soil properties and vegetation. Similarly, Suleiman et al. (2013) reported that land-use history might affect the present-day structure of the soil microbial community, because many highly resilient and resistant microbial populations unrelated to soil type, soil properties, or land use are widely present in the microbial community. Our results showed that the land-use types have nearly the same dominant phyla, genera, and species, although the relative abundances of the taxa were different from one another. For this reason, the influences of land-use conversion were suggested to have more quantitative effects than qualitative effects. In other words, the structure of the soil microbial community may be affected by the changes in the relative abundances of most specific groups. The structure may change in response to land-use conversion, probably because the abundance of specific bacterial taxa sensitive to soil environment variation are extremely low and thus difficult to detect.

The observed phyla are typically present in most soils. Their relative abundances varied among the four land-use types. Hence, land use possibly affected their abundances. *Proteobacteria* and *Acidobacteria* were the most abundant phyla in all of the samples. Proteobacteria are commonly described as abundant free-living bacteria in many habitats (Zhou et al., 2004; Drees et al., 2006; Yang et al., 2015). *Acidobacterial* community is correlated with soil properties (e.g., pH, P, K, Ca, Mg, and Al contents), and its abundance can be altered by soil type and land-use change (Navarrete et al., 2013; Mendes et al., 2015). Our study showed a decrease in the relative abundance of *Acidobacteria* group during the conversion of NMG to UF, PF, and PP. At the genus level, the relative abundance of the dominant genera (*Acidobacteria*, *Blastocatella*, and *Sphingomona*s) also

decreased during this process. This effect indicated that bacterial community structures were altered during land-use conversion.

Our results confirm that land-use conversion directly alters soil properties and microbial community structures in soil. These changes may influence environmental functions. However, the effect of land-use conversion on the functional profile of soil microbes remains unknown. Therefore, further studies on microbial functional profiles in Horqin soils should be performed to understand the function of soil microbes and to evaluate the sustainability of land-use systems.

## 5 Conclusions

1) In this study, land-use conversion in the NMG of the Horqin Sandy Land elicits a distinct effect on the physicochemical parameters of soil and the structure of soil bacterial communities. However, soil type remains the same. Differences among the structures of soil microbial communities are caused by land-use change and variability in land management practices, which directly influence soil physicochemical and microbiological properties.

2) Soil properties, including organic matter, pH, total N, total P, available N and P, and microbial biomass C, N, and P, directly affect microbial community structures. The effects of land-use change on the structure of soil microbial community are mostly quantitative rather than qualitative. For example, their structure is affected by changing the relative abundances of most specific groups, and their structure becomes altered in response to land-use conversion.

3) Proteobacteria, Acidobacteria, Actinobacteria, Bacteroidetes, Chloroflexi, Firmicutes, Verrucomicrobia, Gemmatimonadetes, Planctomycetes, and Nitrospirae are the dominant phyla observed in the soil. Within these phyla, *Acidobacteria*, *Blastocatella*, *Sphingomonas*, and *Bryobacter* are the dominant genera in the Horqin region.

## Acknowledgments

This work was supported by funds from the National Key Research and Development Program of China (Grant

no. 2016YFC0500803) and the National Natural Science foundation of China (41371505). The authors would like to express their gratitude to the members of the Wulanaodu Station of Desertification Research under the Chinese Academy of Sciences for their technical assistance.

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





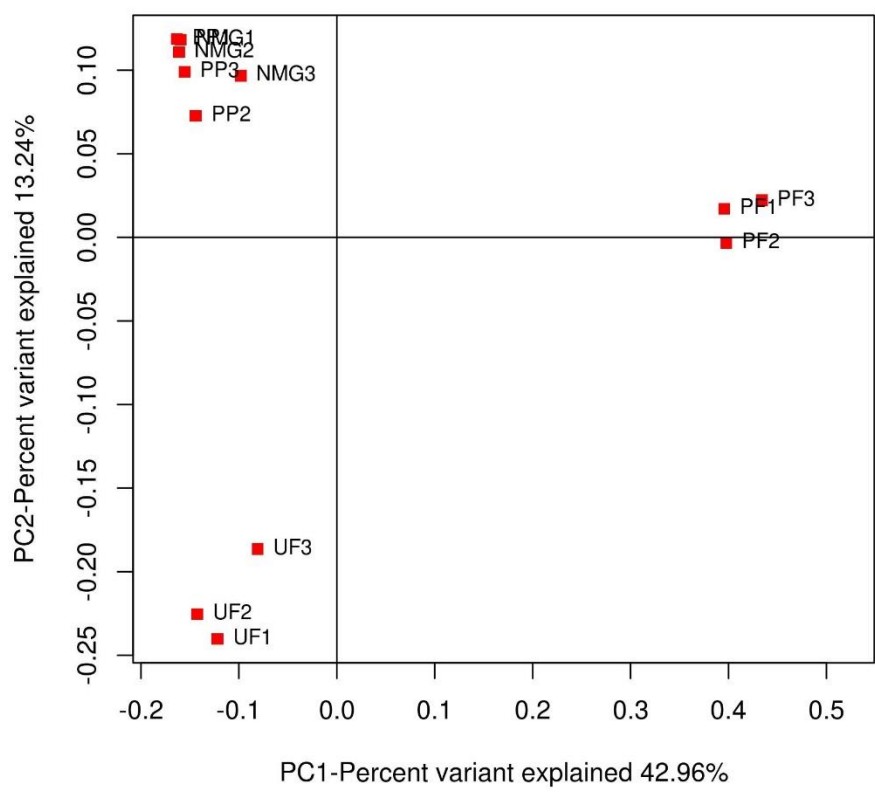

Fig. 1. Ordination plot showing the grouping of soil bacterial communities according to PCoA based on Bray–Curtis distance
matrix.

5      NMG: native meadow grassland; UF: 50-year upland field; PF: 26-year paddy field; PP: 34-year poplar plantation.



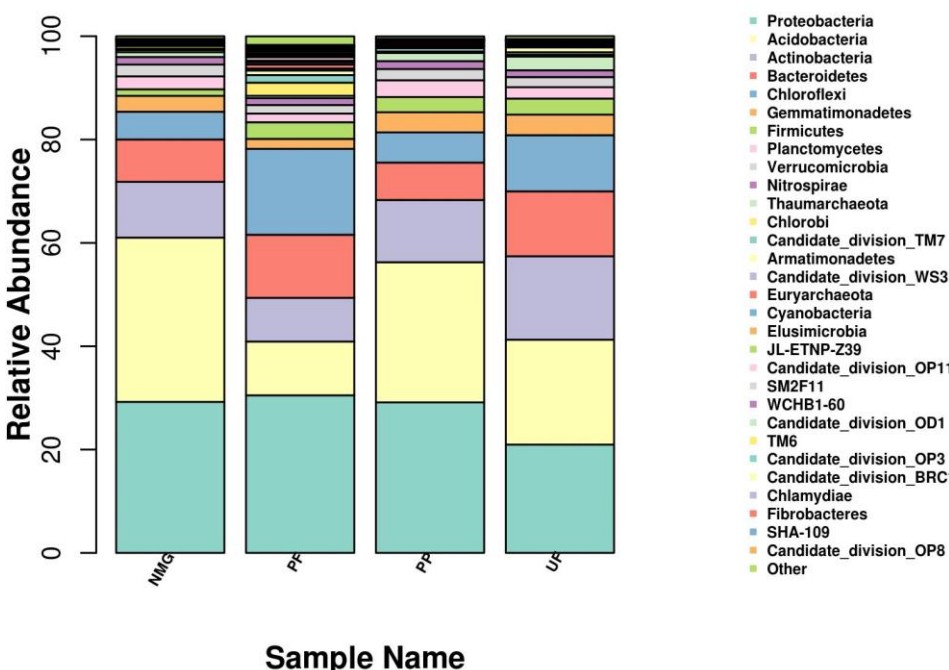

Fig.2. Top taxonomic distribution of soil samples at phylum level.

NMG: native meadow grassland; UF: 50-year upland field; PF: 26-year paddy field; PP: 34-year poplar plantation.



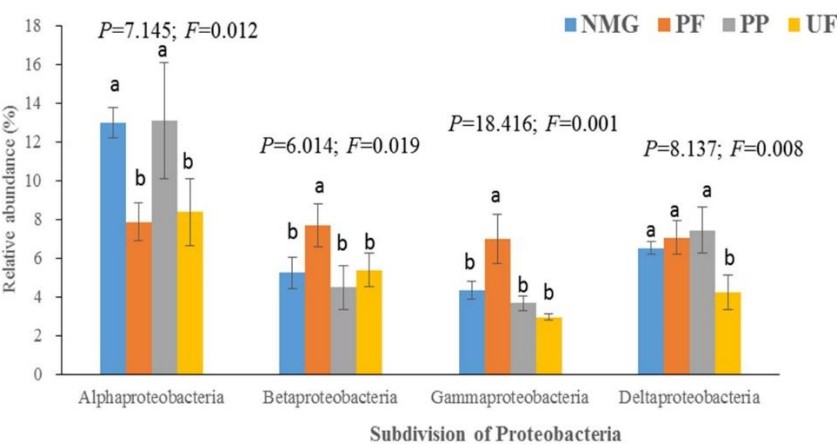

Fig. 3. Class distribution of Proteobacteria phylum.

NMG: native meadow grassland; UF: 50-year upland field; PF: 26-year paddy field; PP: 34-year poplar plantation.



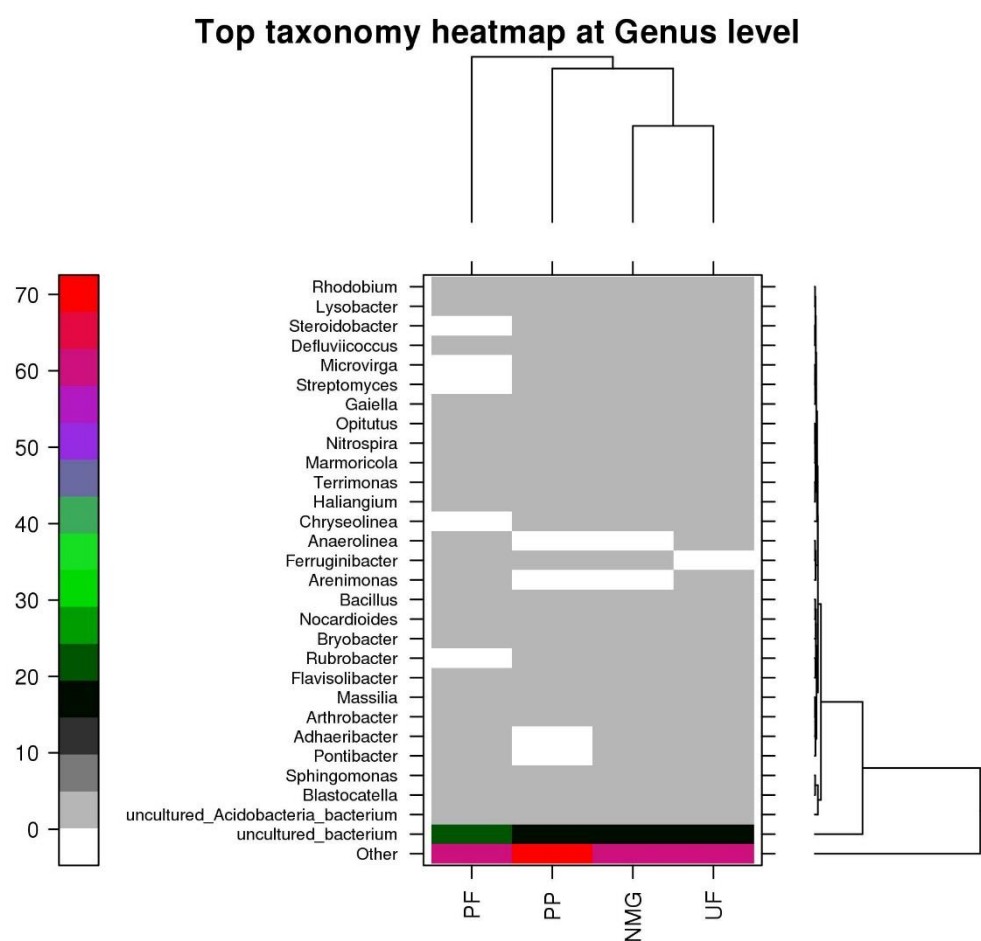

Fig. 4. Top taxonomy heatmap of soil samples at genus level.

NMG: native meadow grassland; UF: 50-year upland field; PF: 26-year paddy field; PP: 34-year poplar plantation.



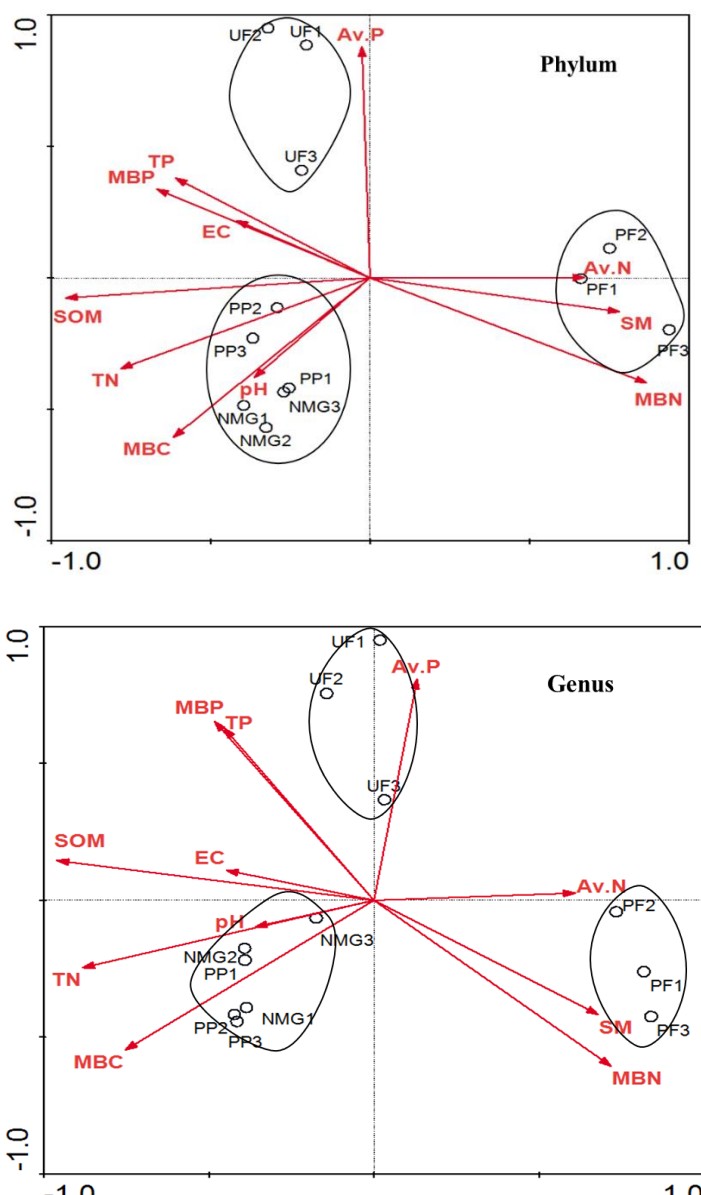

5    **Fig. 5** Canonical correspondence analysis (CCA) of soil bacterial community structures and environmental parameters. Arrows indicate the direction and magnitude of environmental parameters associated with bacterial community structures. Abbreviations: Av. P: available P; SOM: soil organic matter; TP: total P; TN: Total N; EC: electrical conductivity; MBC: microbial biomass C; MBN: microbial biomss N; MBP: microbial biomass P; NMG: native meadow grassland; UF: 50-year upland field; PF: 26-year paddy field; PP: 34-year poplar plantation.



**Table 1**. Soil pH, EC, nutrients, and microbial biomass of samples (average of three replicates ±SD)

| Items | NMG | UF | PF | PP | $P$ | $F$ |
|---|---|---|---|---|---|---|
| Soil moisture (%) | 13.11±2.490a | 13.25±2.632a | 28.26±3.262c | 20.97±1.631b | 23.803 | <0.001 |
| pH | 8.67±0.200b | 7.95±0.128a | 7.85±0.290a | 7.78±0.020a | 14.262 | 0.001 |
| Electrical conductivity ($\mu S.cm^{-1}$) | 86.90±6.329a | 196.0±53.88b | 111.2±33.11a | 306.0±14.91c | 27.631 | <0.001 |
| Organic matter ($g.kg^{-1}$) | 4.186±0.064b | 3.824±0.588b | 1.589±0.130a | 4.579±0.097c | 57.251 | <0.001 |
| Total N ($g.kg^{-1}$) | 0.103±0.005c | 0.084±0.006b | 0.060±0.003a | 0.131±0.003d | 136.790 | <0.001 |
| Total P ($g\ kg^{-1}$) | 0.061±0.007a | 0.073±0.033a | 0.031±0.004a | 0.053±0.006a | 3.109 | 0.089 |
| Available N ($mg.g^{-1}$) | 2.240±0.285a | 1.866±1.090a | 3.219±0.056a | 1.438±0.871a | 3.411 | 0.073 |
| Available P ($mg.g^{-1}$) | 5.252±0.394a | 28.34±11.873b | 12.20±3.034a | 7.457±1.643a | 8.532 | 0.007 |
| Microbial biomass C ($mg.kg^{-1}$) | 317.1±29.638b | 96.84±3.646a | 76.32±2.279a | 466.6±34.71c | 199.019 | <0.001 |
| Microbial biomass N ($mg.kg^{-1}$) | 44.29±3.711b | 17.28±3.014a | 83.40±9.679c | 38.16±2.776b | 73.646 | <0.001 |
| Microbial biomass P ($mg.kg^{-1}$) | 55.43±0.685c | 60.75±1.656d | 19.23±0.698a | 26.68±2.524b | 507.274 | <0.001 |

NMG: native meadow grassland; UF: 50-year upland field; PF: 26-year paddy field; PP: 34-year poplar plantation.

5      Means in row followed by the different letter are significantly different ($P < 0.05$).



**Table 2.** MiSeq sequencing results and diversity estimates for each sampling site

| Sample | Sequencing results | | Diversity estimates | | | |
|---|---|---|---|---|---|---|
| | Total sequences | Total OTUs | ACE | Chao | Shannon | Coverage |
| NMG1 | 62032 | 7298 | 6600 | 6345 | 10.15 | 0.94 |
| NMG2 | 59445 | 7735 | 6695 | 6491 | 10.39 | 0.94 |
| NMG3 | 52383 | 7691 | 7259 | 6988 | 10.62 | 0.93 |
| Mean value | 57953 | 7575 | 6851b | 6608b | 10.39a | 0.94 |
| UF1 | 40255 | 6114 | 6499 | 6286 | 10.49 | 0.94 |
| UF2 | 53850 | 6825 | 6438 | 6236 | 10.46 | 0.94 |
| UF3 | 58719 | 7401 | 6726 | 6555 | 10.55 | 0.94 |
| Mean value | 50941 | 6780 | 6554b | 6359b | 10.50ab | 0.94 |
| PF1 | 66811 | 8744 | 6137 | 6153 | 10.81 | 0.95 |
| PF2 | 46571 | 7279 | 5999 | 5958 | 10.71 | 0.95 |
| PF3 | 45213 | 6776 | 5342 | 5414 | 10.77 | 0.96 |
| Mean value | 52865 | 7600 | 5826a | 5842a | 10.76b | 0.95 |
| PP1 | 57894 | 7579 | 6515 | 6442 | 10.57 | 0.94 |
| PP2 | 62513 | 7731 | 6771 | 6561 | 10.31 | 0.94 |
| PP3 | 40441 | 6255 | 6860 | 6605 | 10.36 | 0.94 |
| Mean value | 53616 | 7188 | 6715b | 6536b | 10.41a | 0.94 |
| *F* | - | - | 6.912 | 4.836 | 4.497 | - |
| *P* | - | - | 0.013 | 0.033 | 0.040 | - |

ACE: abundance-based coverage estimator, Chao: Chao's species richness estimator, Shannon: Shannon-Weiner Index.

Species level, 97% similarity threshold used to define the OTUs

NMG: native meadow grassland; UF: 50-year upland field; PF: 26-year paddy field; PP: 34-year poplar plantation.

Means in column followed by the different letter are significantly different ($P < 0.05$).





**Table 3.** Results of one way ANOVA for the relative abundance of phylum and genus at different land-use types

| Order | Phylum | $F$ | $P$ | Genus | $F$ | $P$ |
|---|---|---|---|---|---|---|
| 1 | *Proteobacteria* | 7.58 | 0.010 | uncultured_bacterium | 21.03 | <0.001 |
| 2 | *Acidobacteria* | 24.47 | <0.001 | *Acidobacteria*_bacterium | 13.60 | 0.002 |
| 3 | *Actinobacteria* | 14.55 | 0.001 | *Blastocatella* | 14.98 | 0.001 |
| 4 | *Bacteroidetes* | 4.817 | 0.034 | *Sphingomonas* | 2.843 | 0.105 |
| 5 | *Chloroflexi* | 21.33 | <0.001 | *Bryobacter* | 2.338 | 0.150 |
| 6 | *Gemmatimonadetes* | 21.38 | <0.001 | *Arthrobacter* | 9.194 | 0.006 |
| 7 | *Firmicutes* | 2.219 | 0.163 | *Massilia* | 11.92 | 0.003 |
| 8 | *Planctomycetes* | 19.09 | 0.001 | *Nocardioides* | 3.937 | 0.054 |
| 9 | *Verrucomicrobia* | 1.330 | 0.331 | *Adhaeribacter* | 7.012 | 0.013 |
| 10 | *Thaumarchaeota* | 6.851 | 0.013 | *Flavisolibacter* | 7.824 | 0.009 |
| 11 | *Nitrospirae* | 0.288 | 0.833 | *Chryseolinea* | 30.80 | <0.001 |
| 12 | *Chlorobi* | 16.60 | 0.001 | *Rubrobacter* | 4.162 | 0.047 |
| 13 | Candidate_division_TM7 | 18.91 | 0.001 | *Pontibacter* | 8.728 | 0.007 |
| 14 | *Armatimonadetes* | 30.36 | <0.001 | *Bacillus* | 1.616 | 0.261 |
| 15 | *Candidate_division_WS3* | 0.890 | 0.486 | *Nitrospira* | 5.724 | 0.022 |
| 16 | *Cyanobacteria* | 6.239 | 0.017 | *Haliangium* | 1.305 | 0.338 |
| 17 | *Euryarchaeota* | 285.6 | <0.001 | *Gaiella* | 9.340 | 0.005 |
| 18 | *Elusimicrobia* | 2.645 | 0.121 | *Marmoricola* | 2.819 | 0.107 |
| 19 | JL-ETNP-Z39 | 3.748 | 0.060 | *Arenimonas* | 8.997 | 0.006 |
| 20 | Candidate_division_OP11 | 364.2 | <0.001 | *Streptomyces* | 7.261 | 0.011 |

Phylum and genus are arranged in order according to the relative abundances

