# Peer review of "Land-use changes influence soil bacterial communities in a meadow grassland in Northeast China"

_Solid Earth, 2017_

## Referee Comment (RC1) · Anonymous Referee #1 · 7 Aug 2017

General comments: The MS presented by Cao et al, investigated the effects of land-use changes on soil properties and bacterial communities in Horqin Grasslands of Northeast China. The shifts in the diversities and compositions of soil bacterial communities and the relative abundance of dominant taxa were detected. Experimental design and methodology were well executed and the results were professionally analyzed. The experimental set-up was appropriate: Four land-use systems (natural meadow grassland, paddy field, upland field, and poplar plantation) were selected as experimental sites. Composite samples were taken from three random field replicates established at each site. Also the methods used in soil properties analysis seem adequate. Especially, high-throughput Illumina MiSeq sequencing technique was applied

in this paper to investigate the changes in the microbial diversities and community compositions, which can provide useful information for microbial community composition, therefore, the results is credible. The results will be useful, in that it gives some proofs for land use management in semi-arid eroded environment. The paper is within the scope of SE, and the research was performed in an interesting area of China. Generally, the MS is easy to read. The content and form of this article is well focused.

Specific comments - Vegetation flora classifications should cite references -Provide authorship for plant species -Change "physicochemical" to "physical and chemical" soil properties here and elsewhere - Please check acronyms. For nonstandard acronyms and initialisms that appear more than once in the main text, that is, they are usually defined first as per common usage in the field of study, provide the full term at first mention, with the acronym or initialism indicated in parentheses. Thereafter, use the acronym or initialism alone in subsequent mentions. -P8 L11 Verify if large numbers should be properly inserted with commas for clarity. -P11 L4 "Land-use conversion can be regarded as the determinant that can alter soil properties because the four sites have the same soil type." : not clear. -P14 L4 Instead of saying "These changes may influence environmental functions", it is better to say "These changes in soil properties and microbial community structures may produce potential effects on the ecological functions of the meadow grassland ecosystem." -P14 L19-22, redundant conclusion, delete 3)...... - Fig. 3 change "class" to "subdivision"

---

## Author Comment (AC1) · 10 Aug 2017

Thanks for your kindly comments. As suggested, we will add the references for vegetation description and the authorship for plant species, check acronyms all text, and accept all your modification suggestions in Specific Comments.

---

## Referee Comment (RC2) · Anonymous Referee #2 · 20 Aug 2017

General comments: In this study, authors investigated the effects of land-use changes on soil physicochemical properties and bacterial communities in a semi-arid region in Northeast China. The diversity and composition of soil bacterial community and the relative abundance of dominant taxa were detected in four land-use types (natural grassland, paddy field, upland field, and poplar plantation) through high-throughput Illumina MiSeq sequencing technique. The results provide useful information for microbial community composition under land-use changes in semi-arid environment. However, I have two major concerns: (1) In a given site, land-use type, land-use history (including duration), and land management practices (such as fertilization) are expected to impact the soil bacterial community by altering nutrient availability and environmental

conditions. In this study, you have four land-use types: native meadow, 26-year paddy field, 50-year upland field, and 34-year poplar plantation (the three latter ones were all converted from the former one; with different use durations). You should discuss the possible effects of duration in the Discussion section (Page 13, L7-11) (2) Another concern is the sampling time. The soils were sampled in May 2016. You should add explanations why sampling in May is rational given that seasonal changes exist in many variables measured in this study?

Specific comments: Page 2, L3, change 'are converted' to 'have been converted' Page 2, L4, the meaning of 'response processes' is ambiguous. Page 2, L5, 'ecosystem services' is better than 'environmental services of ecosystems' Page 2, L9-10, the statement of this sentence (In the coming decades, many semi-arid grassland areas in Northeast China are expected to be converted for agriculture use) needs evidence to support. Page 2, L11, delete 'influencing' after 'key' Page 2, L24, Change 'have yet to be sufficiently assessed' to 'have not yet been sufficiently assessed' Page 4, L6, 'the effects of native grassland cultivation'? How did you cultivate the native grassland? Page 4, L7, change 'the future development of sustainable use of fragile ecosystem services' to 'the sustainable use of ecosystem services of fragile grasslands' Page 4, L7-10, delete '(namely, native meadow grassland, upland field, paddy field, and woodland)' Page 4, L20, Horqin Grassland? Is it a specific type of grassland or a name for a specific location? Page 5, L7, 'The meadow grassland is' Page 7 L11-12, you defined the abbr 'AEC' (abundance-based coverage estimator), but you used 'evenness (ACE)' again in Page 8, L14. Please clarify the abbr. Page 10, L11, '3.5 Relationship between microbial community composition and soil environmental factors' Page 14, L4, 'These changes may influence environmental functions'. What do environmental functions mean here? Page 14, L10-11 delete "However, soil type remains the same." It makes no sense. Page 16, L8, add a comma (,) after 'by pH'

---

## Author Comment (AC2) · 21 Aug 2017

We thank reviewer's valuable comments and suggestions. We will revise the manuscript as you suggested, especially your two major concerns and the modification suggestions in Specific Comments.
* * *

---

## Author Response (AR1)

**Response to Referees' and Editor's Comments**

We thank the editor for giving us the opportunity to revise, and thank reviewers' valuable comments and suggestions, and reply them one by one as follows:

**Reply to referee 1:**

**Q1: Vegetation flora classifications should cite references –Provide authorship for plant species.**

- Done as suggested. The authorship for plant species have been added in the "2.1 Study location and site description", and a reference (Cao et al., 2008) has been cited as well.

**Q2: Change "physicochemical" to "physical and chemical" soil properties here and elsewhere.**

- Done as suggested. All "physicochemical" in the text have been replaced by physical and chemical.

**Q3: Please check acronyms. For nonstandard acronyms and initialisms that appear more than once in the main text, that is, they are usually defined first as per common usage in the field of study, provide the full term at first mention, with the acronym or initialism indicated in parentheses. Thereafter, use the acronym or initialism alone in subsequent mentions.**

- Done as suggested. We have rechecked all of the acronyms in the text.

**Q4: P8 L11 Verify if large numbers should be properly inserted with commas for clarity.**

- Done as suggested. Commas have been inserted in some large numbers.

**Q5: P11 L4 "Land-use conversion can be regarded as the determinant that can alter soil properties because the four sites have the same soil type." not clear.**

- The sentence has been deleted.

**Q6: P14 L4 Instead of saying "These changes may influence environmental functions", it is better to say "These changes in soil properties and microbial community structures may produce potential effects on the ecological functions of the meadow grassland ecosystem."**

- The sentence has been replaced as suggested.

**Q7: -P14 L19-22, redundant conclusion, delete 3):**

- The paragraph has been deleted.

**Q8: - Fig. 3 change "class" to "subdivision"**

    - Done as suggested.

**Reply to referee 2:**

General comments:

Q1: You should discuss the possible effects of duration in the Discussion section (Page 13, L7-11)

    - Done as suggested. Several relative sentences have been added.

Q2: The soils were sampled in May 2016. You should add explanations why sampling in May is rational given that seasonal changes exist in many variables measured in this study?

    - Indeed, soil properties seasonally change with vegetation growth and environmental variation, however, in this study we focused on the bacterial community response to land-use conversion of grassland and the correlation with soil properties at the time of sampling. All samples were taken at the same time (May). We did not concern the dynamics of microbial community and soil properties in this study.

Specific comments:

Q1: Page 2, L3, change 'are converted' to 'have been converted'

    - Done as suggested.

Q2: Page 2, L4, the meaning of 'response processes' is ambiguous.

    - It has been modified as "ecological processes of the conversions"

Q3: Page 2, L5, 'ecosystem services' is better than 'environmental services of ecosystems'

    - Done as suggested.

Q4: Page 2, L9-10, the statement of this sentence (In the coming decades, many semi-arid grassland areas in Northeast China are expected to be converted for agriculture use) needs evidence to support.

    - The sentence has been deleted.

Q5: Page 2, L11, delete 'influencing' after 'key'

    -Done as suggested.

Q6: Page 2, L24, Change 'have yet to be sufficiently assessed' to 'have not yet been sufficiently assessed'

-Done as suggested.

Q7: Page 4, L6, 'the effects of native grassland cultivation'? How did you cultivate the native grassland?

   - It has been revised as "the effects of use change of native grassland"

Q8: Page 4, L7, change 'the future development of sustainable use of fragile ecosystem services' to 'the sustainable use of ecosystem services of fragile grasslands'

   - Done as suggested.

Q9: Page 4, L7-10, delete '(namely, native meadow grassland, upland field, paddy field, and woodland)'

   - Done as suggested.

Q10: Page 4, L20, Horqin Grassland? Is it a specific type of grassland or a name for a specific location?

   - It is a specific location, thus "Horqin Grassland" has been revised as "Horqin grassland" here and elsewhere.

Q11: Page 5, L7, 'The meadow grassland is'

   - Done as suggested.

Q12: Page 7 L11-12, you defined the abbr 'AEC' (abundance-based coverage estimator), but you used 'evenness (ACE)' again in Page 8, L14. Please clarify the abbr.

   - Done as suggested. We have rechecked all of the acronyms in the text and have modified the spelling mistakes.

Q13: Page 10, L11, '3.5 Relationship between microbial community composition and soil environmental factors'

   - Done as suggested.

Q14: Page 14, L4, 'These changes may influence environmental functions'. What do environmental functions mean here?

   - As suggested by referee 1, the sentence has been modified as "These changes may produce potential effects on the ecological functions of the meadow grassland ecosystem".

Q15: Page 14, L10-11 delete "However, soil type remains the same." It makes no sense.

   - The sentence has been removed.

Q16: Page 16, L8, add a comma (,) after 'by pH'

   - Done as suggested.

[revised manuscript text omitted]